# Associations between corticosteroid dosage and clinical outcomes in patients with hypoxemic COVID-19 pneumonia: A retrospective cohort study

**Napassorn Teeratakulpisarn**, **Supattra Chiewroongroj, Thummaporn Naorungroj, Ranistha Ratanarat** *

Department of Medicine, Faculty of Medicine Siriraj Hospital, Mahidol University, Bangkok, Thailand

* ranittha@hotmail.com

## Abstract

### Background

Corticosteroids are commonly used to treat COVID-19 patients with hypoxemia, and clinicians have adjusted the corticosteroid intensity on the basis of clinical needs. However, neither the optimal dose nor the duration of treatment has been recommended.

### Objective

To investigate whether cumulative doses of corticosteroids, measured as dexamethasone-equivalent doses over the first 14 days, impact outcomes in patients with COVID-19 pneumonia.

### Methods

We conducted a retrospective cohort study of COVID-19 pneumonia patients admitted between April 1st, 2020, and September 30th, 2021. The study focused on the type and dose of corticosteroid administered during the initial 14 days, clinical outcomes, and complications. The primary outcome was in-hospital mortality.

### Results

Among 271 patients, the mean cumulative dexamethasone-equivalent dose was 158 (119.9–197.25) mg in survivors and 185 (131.7–222.0) mg in nonsurvivors. Univariate analysis revealed that the cumulative dexamethasone-equivalent dose was a risk factor for in-hospital mortality. However, this association did not hold true in the multivariate analysis. After the cumulative dexamethasone-equivalent dose was categorized into quartiles, the moderate dosage (126.01–165.00 mg) in the second quartile was found to be associated with the lowest in-hospital mortality (16.2%). Higher cumulative dexamethasone-equivalent doses were associated with longer hospital and ICU stays and fewer ventilator-free days (p

**Data Availability Statement:** All relevant data are within the manuscript and its Supporting Information files.

**Funding:** The author(s) received no specific funding for this work.

**Competing interests:** The authors have declared that no competing interests exist.

< 0.001). Doses exceeding 165 mg were associated with an increased risk of hospital-acquired infections (p < 0.001).

## Conclusions

The cumulative dexamethasone-equivalent dose during the first 14 days is not associated with in-hospital mortality in hypoxemic COVID-19 patients. However, higher cumulative doses exceeding 165 mg are associated with an increased risk of in-hospital mortality and secondary hospital-acquired infections.

## Introduction

COVID-19 is an infectious disease caused by severe acute respiratory syndrome coronavirus 2 (SARS-CoV-2), which emerged in late 2019 and was declared a pandemic by the World Health Organization (WHO) in March 2020 [1]. The severity of the disease ranges from asymptomatic to severe pneumonia with acute respiratory distress syndrome and fatal outcomes, particularly in patients with comorbidities, who have a greater potential risk of severe COVID-19 [2]. The pathogenesis of COVID-19 involves two main processes: an early phase driven by SARS-CoV-2 replication and a later phase characterized by immune dysregulation and an inflammatory response, leading to tissue damage. Early-phase treatment primarily involves antiviral therapy, whereas immunosuppressive and anti-inflammatory therapies are more beneficial in the later stage [3].

Immunomodulatory therapy for COVID-19 mainly relies on corticosteroids, particularly dexamethasone. However, dexamethasone has inconsistent outcomes depending on oxygen dependence. Landmark studies, such as the Randomized Evaluation of COVID-19 Therapy (RECOVERY) trial and the COVID-19 Dexamethasone (CODEX) trial, demonstrated that the use of dexamethasone showed benefits of mitigating disease severity and death in COVID-19 patients receiving oxygen therapy [4, 5]. Conversely, a systematic review and meta-analysis revealed that the use of dexamethasone in patients not requiring oxygen was linked to increased disease severity and mortality risk [6]. Moreover, studies using alternative corticosteroids (e.g., methylprednisolone and hydrocortisone) in COVID-19 patients reported no improvement in clinical outcomes or mortality rates [7–10]. Consequently, the current National Institute's COVID-19 treatment guidelines recommend the use of dexamethasone only in patients requiring oxygen therapy [3]. Although one study did compare high- vs. low-dose corticosteroids, it did not report statistically significant outcomes [11]. A consensus on the optimal dose and duration of dexamethasone treatment is still needed.

Therefore, we investigated whether corticosteroid intensity, measured by the cumulative dexamethasone-equivalent dose over the first 14 days, affects in-hospital mortality in patients with COVID-19 pneumonia.

## Materials and methods

### Study design

This single-center retrospective observational study was conducted at Siriraj Hospital, a tertiary hospital affiliated with Mahidol University, Thailand, from April 1st, 2020, to September 30th, 2021. The study protocol was authorized by the Siriraj Institution Review Board (approval no. Si-355/2020).

## Participant selection

We included all COVID-19 patients over 18 years of age with pneumonia who required oxygen therapy and were admitted to Siriraj Hospital. The COVID-19 diagnoses were confirmed by real-time reverse transcription–polymerase chain reaction. Patients who had "do not resuscitate" requests or declined endotracheal intubation were excluded from the study.

## Outcomes

The primary outcome of this study was in-hospital mortality. The secondary outcomes were ICU mortality and 28-day and 90-day mortality.

## Study procedure

We retrospectively reviewed the patients' data, including demographic information, underlying diseases, duration of symptoms, hemodynamic parameters, Sequential Organ Failure Assessment (SOFA) score, and Acute Physiology and Chronic Health Evaluation II (APACHE II) score. We also collected data from laboratory investigations (arterial blood gas analysis, C-reactive protein, interleukin-6, complete blood count, blood chemistry, and liver function tests). Written informed consent was waived due to the retrospective study design, and approval was obtained from the Siriraj Institution Review Board (approval no. Si-355/2020). Data were accessed for research purposes from January 24, 2023, to April 15, 2023.

We recorded treatment interventions administered during hospitalization, such as antiviral drugs, corticosteroids, other anti-inflammatory agents (e.g., tocilizumab and baricitinib), hemoperfusion therapy, oxygen therapy, prone positioning, renal replacement therapy, and extracorporeal membrane oxygenation (ECMO) therapy provided by nurses, according to the physician's order. The type and daily dose of corticosteroids given to each patient were recorded for the first 14 days and expressed as the dexamethasone-equivalent dose. The cumulative dexamethasone-equivalent dose during the initial 14 days was used to determine the intensity of corticosteroid treatment. We documented concurrent oxygen therapies administered during the same period. The table for calculating the dexamethasone-equivalent dose is shown in **S1 Table**. In addition, we reviewed potential complications associated with corticosteroid therapy, including hospital-acquired infections including hospital-acquired pneumonia, urinary tract infection, bacteremia, catheter-related bloodstream infection, skin and soft tissue infection, and gastrointestinal bleeding.

## Statistical analyses

Normality tests were conducted for qualitative data. Normally distributed data are expressed herein as means and standard deviations, whereas nonnormally distributed data are presented as medians and interquartile ranges (IQRs). Categorical data are reported as frequencies and percentages. To identify risk factors significantly correlated with mortality outcome, comparisons between survivors and nonsurvivors were made via the chi-square test, Fisher's exact test, Student's t test, and the Mann–Whitney U test when appropriate. Variables with $p < 0.05$ in the univariate analysis were included in a multivariate analysis via a multiple logistic regression model. The cumulative dose of dexamethasone received by each patient in the first 14 days was calculated and divided into quartiles (Q1-Q4). Mortality outcomes and complications were compared via the chi-square test. Kruskal–Wallis tests were used to analyze other clinical outcomes. Data analyses were performed via IBM SPSS Statistics, version 29 (IBM Corp, Armonk, NY, USA). Two-sided p values $< 0.05$ were considered statistically significant.

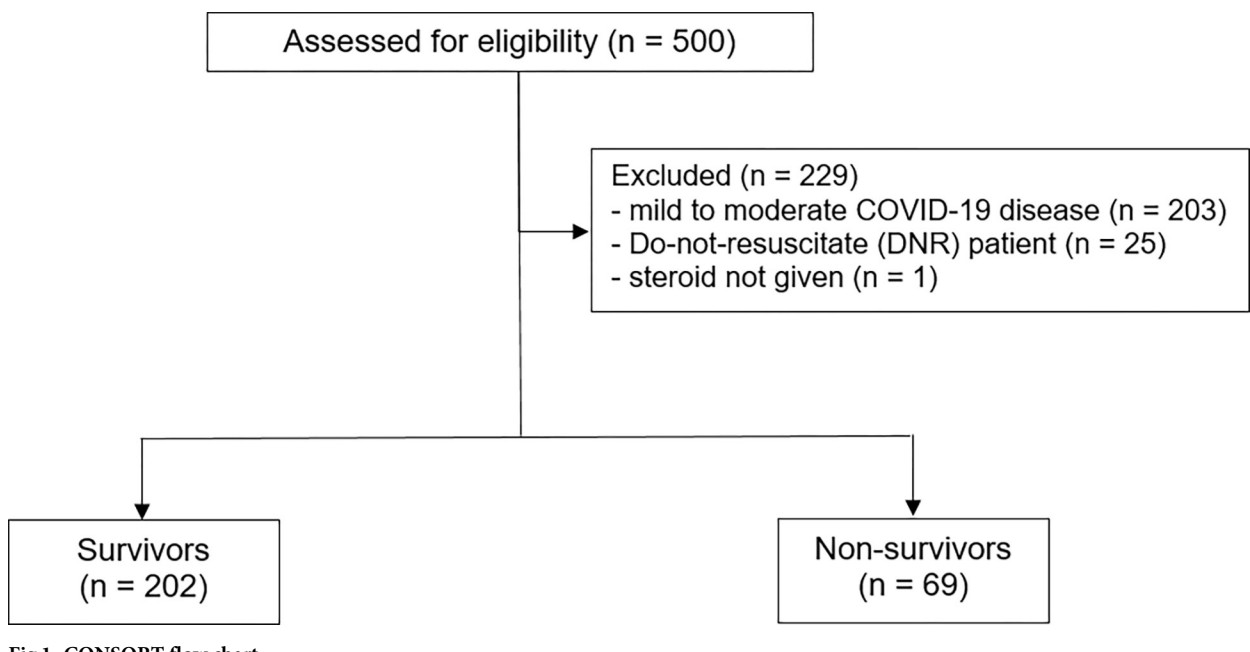

**Fig 1. CONSORT flow chart.**

## Results

### Baseline characteristics

Data were collected from 271 patients with COVID-19 pneumonia who required oxygen therapy at Siriraj Hospital between April 1st, 2020, and September 30th, 2021. Among these, 202 (74.5%) were in the survivor group, and 69 (25.5%) were in the nonsurvivor group (Fig 1). The characteristics of the study patients and treatment according to in-hospital mortality are summarized in **Table 1**. In the overall cohort, the median age of the patients was 62 years (IQR: 51, 69). Most patients were male, accounting for 66% of the study population. The baseline APACHE II and SOFA scores were 10 (IQR: 7, 13) and 3 (IQR: 2, 4), respectively. All patients had hypoxemia, with a median $PaO_2/FiO_2$ (PF) ratio of 151.4 (IQR: 104.8, 224.0). The median cumulative dexamethasone-equivalent dose administered in the first 14 days was 165 mg (IQR: 126, 203). In-hospital mortality occurred in 69 patients (25.5%). Survivors were younger, had a lower proportion of chronic kidney disease, and had lower severity scores (SOFA and APACHE II scores). In contrast, nonsurvivors had lower lymphocyte counts and PF ratios. Compared with survivors, nonsurvivors had a more significant dose of cumulative dexamethasone equivalence, received ECMO and mechanical ventilation more frequently, and had a longer duration of total ventilator days (**Table 1**).

### Outcomes

In univariate analyses, risk factors, including age, chronic kidney disease, baseline SOFA score, lymphocyte count, baseline PF ratio, cumulative dexamethasone dose in the first 14 days, and ventilator requirement, were significantly associated with in-hospital mortality (**Table 1**). When these factors were included in the multivariable analysis, the risk factors that remained independently associated with in-hospital mortality were age, chronic kidney disease, baseline PF ratio, and ventilator requirement but not the cumulative dexamethasone dose (**S2 Table**). Although the cumulative dexamethasone equivalent dose was not statistically significant in the multivariable analysis, it remained a potentially modifiable risk factor during treatment.

**Table 1. Baseline characteristics of the patients and treatment-related data according to in-hospital mortality.**

| variables | Survivors | Nonsurvivors | p value |
|---|---|---|---|
| | (n = 202) | (n = 69) | |
| Male, no. (%) | 130 (64%) | 50 (73%) | 0.22 |
| Age, years | 57.82 ± 15.79 | 67.04 ± 12.28 | < .001 |
| Body mass index, kg/m$^2$ | 27.45 ± 6.63 | 27.37 ± 6.04 | 0.93 |
| Underlying diseases, no. (%) | | | |
| • Obesity (BMI > 30) | 56 (27.7%) | 22 (31.9%) | 0.51 |
| • Diabetes mellitus | 90 (44.6%) | 35 (50.7%) | 0.38 |
| • Hypertension | 110 (54.4%) | 49 (71%) | 0.02 |
| • COPD | 5 (2.5%) | 4 (5.8%) | 0.18 |
| • Coronary artery disease | 19 (9.4%) | 11 (15.9%) | 0.14 |
| • Chronic kidney disease | 31 (15.3%) | 22 (31.9%) | 0.003 |
| • Immunosuppression | 10 (5%) | 5 (7.4%) | 0.46 |
| APACHE II | 9.5 ± 4.8 | 12.2 ± 4.9 | 0.001 |
| SOFA | 2.5 (2–4) | 3 (2–5) | < 0.001 |
| Laboratory investigations at baseline | | | |
| • C-reactive protein level, mg/L | 73.2 ± 64 | 80.6 ± 79 | 0.44 |
| • Lymphocyte count, cell/mm$^3$ | 521.8 (303.7–960.2) | 406.1 (287.3–660.5) | 0.03 |
| • Serum interleukin-6 level, pg/ml | 20.8 (10.5–77.9) | 21.0 (12.3–120.6) | 0.1 |
| • Serum procalcitonin level, ng/ml | 0.2 (0.1–0.9) | 0.2 (0.1–1.0) | 0.10 |
| • PaO2/FiO2 ratio | 143.3 (112.6–178.6) | 141 (99.3–185.4) | < .001 |
| Treatment | | | |
| Cumulative dexamethasone-equivalent dose during D1-D14, mg | 158 (119.9–197.25) | 185 (131.7–222.0) | 0.02 |
| DOS before steroid treatment, day | 5 (3–8) | 4 (3–5.5) | 0.43 |
| Remdesivir, no. (%) | 57 (28.2%) | 28 (40.6%) | 0.06 |
| Tocilizumab, no. (%) | 32 (15.8%) | 16 (23.2%) | 0.17 |
| Baricitinib, no. (%) | 7 (3.5%) | 3 (4.3%) | 0.74 |
| Cytokine adsorptive therapy, no. (%) | 29 (14.4%) | 15 (21.7%) | 0.15 |
| ECMO, no. (%) | 1 (0.5%) | 6 (8.7%) | < 0.001 |
| Ventilator requirement, no. (%) | 78 (38.6%) | 67 (97.1%) | < 0.001 |
| Total ventilator days, days | 5.5 (0–11.5) | 8.5 (8–33.5) | < 0.001 |

Normally distributed variables are presented as the mean ± standard variation; nonnormally distributed variables are presented as the median (interquartile range). Abbreviations: APACHE II, Acute Physiology and Chronic Health Evaluation II; BMI, body mass index; IQR, interquartile range; PaO2/FiO2 ratio: the ratio of arterial oxygen partial pressure to fractional inspired oxygen; SD: standard difference; SOFA: sequential organ failure assessment score; D1-D14, days since steroid therapy was initiated; DOS, day of symptom onset; ECMO, extracorporeal membrane oxygenation therapy; IQR, interquartile range.

Therefore, patients were divided into quartiles on the basis of the cumulative dexamethasone dose in the first 14 days (8–126.00, 126.01–165.00, 165.01–203.00, and 203.01–481.40 mg) (**S3 Table**). The second quartile group (126.01–165.00 mg/14 days) had the lowest in-hospital mortality rate at 16.2%, followed by the first, third, and fourth quartile groups at 22.1%, 25.0%, and 38.8%, respectively (**Fig 2**). The ICU mortality, 28-day mortality, and 90-day mortality rates showed similar patterns with statistical significance. Details of mortality and other clinical outcomes are presented in **Table 2**.

## Complications

The complications related to corticosteroids are shown in **Table 2**. Higher cumulative dexamethasone doses were significantly associated with an increased incidence of hospital-

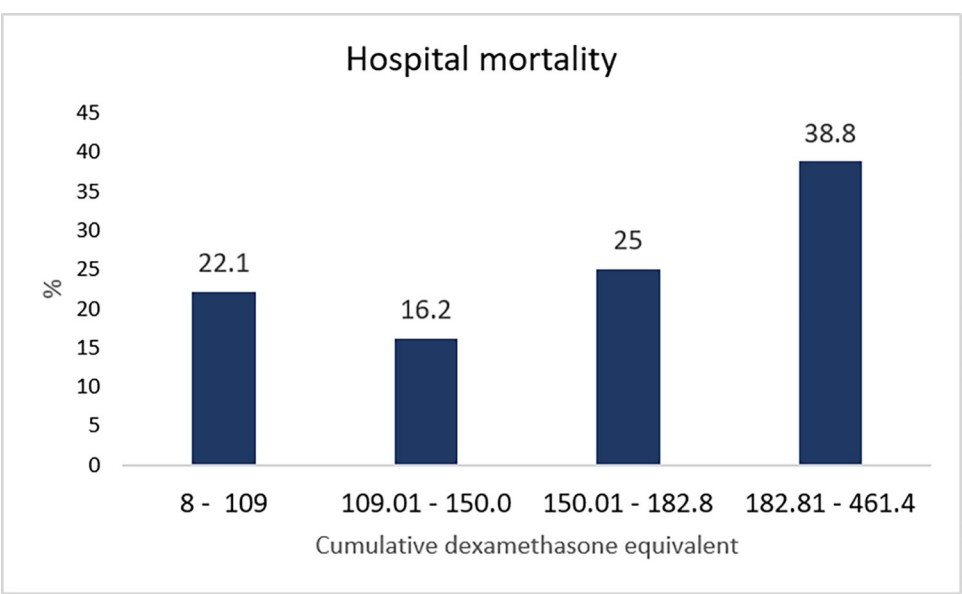

**Fig 2. Hospital mortality based on the cumulative dose of dexamethasone.** [a] statistically significant with
$p < 0.005$.

acquired infections ($p < 0.001$) and tended to increase the occurrence of gastrointestinal
bleeding (p = 0.06).

## Discussion

In this retrospective cohort study of COVID-19 pneumonia patients, multivariate analysis
revealed that the cumulative dexamethasone-equivalent dose was not a risk factor associated

**Table 2. Clinical outcomes stratified into quartiles according to the cumulative dexamethasone-equivalent dose in the first 14 days.**

| Clinical outcomes | Cumulative dexamethasone-equivalent dose (mg) | | | | |
|---|---|---|---|---|---|
| | 8–126.00 | 126.01–165.00 | 165.01–203.00 | 203.01–481.4 | *p*—value |
| | [Q1] | [Q2] | [Q3] | [Q4] | |
| ICU mortality, no. (%) | 10 (14.7%) | 6 (8.8%) | 12 (17.6%) | 18 (26.9%) | 0.01 |
| In-hospital mortality, no. (%) | 15 (22.1%) | 11 (16.2%) | 17 (25.0%) | 26 (38.8%) | 0.004 |
| 28-day mortality, no. (%) | 14 (24.1%) | 7 (11.5%) | 10 (16.4%) | 15 (23.4%) | 0.29 |
| 90-day mortality, no. (%) | 16 (29.1%) | 12 (21.1%) | 18 (32.1%) | 26 (42.6%) | 0.02 |
| ICU LOS, days | 8.5 (6–12) | 10 (6.25–15.75) | 11.50 (7–16.75) | 16 (8–24) | < 0.001 [a,b] |
| Hospital LOS, days | 13 (9–17) | 15 (11.25–24) | 20 (14–31.75) | 26 (17–40) | < 0.001 [a,b,c] |
| Total ventilator days, days | 7.5 (3.25–12.75) | 10.5 (7–19.25) | 9 (5.5–17.75) | 20 (8–34) | 0.003 [a] |
| Ventilator-free days at day 28, days | 28 (20.75–28) | 27 (17.5–28) | 23 (1.25–28) | 2 (0–28) | < 0.001 [a,b,c,d] |
| Hospital acquired infection, no. (%) | 27 (39.7%) | 38 (55.9%) | 45 (66.2%) | 52 (77.6%) | < 0.001 |
| GI bleeding, no. (%) | 7 (10.3%) | 8 (11.8%) | 7 (10.3%) | 16 (23.9%) | 0.06 |

Abbreviations: ICU, intensive care unit; IQR, interquartile range; LOS, length of stay; GI, gastrointestinal; Q, quartile

[a] Pairwise comparison revealed significant data with asymptotic significance (2-sided tests), with $p < 0.05$ between Q1 and Q4.

[b] Pairwise comparison revealed significant data with asymptotic significance (2-sided tests), with $p < 0.05$ between Q2 and Q4.

[c] Pairwise comparison revealed significant data with asymptotic significance (2-sided tests), with $p < 0.05$ between Q1 and Q3.

[d] Pairwise comparison revealed significant data with asymptotic significance (2-sided tests), with $p < 0.05$ between Q3 and QQ.

with increased in-hospital mortality. However, in the subgroup analysis, when we divided the cumulative dexamethasone-equivalent doses into quartiles, the second quartile (P25–P50; 126.01–165.00 mg/14 days) was significantly associated with the lowest mortality. Moreover, a greater cumulative dexamethasone-equivalent dose was associated with increased ICU LOS, hospital LOS and ventilator-free days at day 28.

The recommended dose and regimen of corticosteroids in COVID-19 pneumonia patients with hypoxemia remain inconclusive. Studies such as the COVID STEROID2 study and those conducted by Maskin LP et al. have shown trends favoring higher doses of dexamethasone, with benefits in days alive without life support and time required for cessation of mechanical ventilation [11, 12].

However, conflicting results have been reported, such as those of the randomized controlled trial by Toroghi et al., who found no clinical benefit of high-dose dexamethasone over conventional low-dose dexamethasone [13]. Similarly, the COVIDICUS trial did not demonstrate improved 60-day survival with a higher cumulative dexamethasone dose [14]. These discrepancies may be attributed to differences in corticosteroid intensity and disease severity across the various studies, as shown in **S4 Table**.

Our findings demonstrated the relationship between increased intensity of corticosteroids administered daccording to the hypoxemia severity and improved in-hospital mortality. This association was shown between the lowest intensity (dexamethasone equivalent < 126 mg) in patients with a median PF ratio of 290 (190–439) and a moderate intensity (126.01–165.00 mg) with a median PF ratio of 175 (129–268) (**Table 2** and **Fig 2**). However, this association disappeared in patients with a greater degree of hypoxemia. Increasing the cumulative dexamethasone-equivalent dose beyond 165 mg was associated with increased in-hospital mortality.

This phenomenon could be explained by the fact that higher doses of steroids (> 165 mg) lead to an increased risk of hospital-acquired infections, as shown in **Table 2**. The incidence of secondary infections increased proportionally with an increasing cumulative dexamethasone dose, with statistical significance observed.

This "J-shaped" pattern aligns with the results of a retrospective study conducted by Maia et al. [15], which showed a strong correlation between the cumulative corticosteroid dose (expressed as a methylprednisolone equivalent) and the duration of mechanical ventilation. The shortest ventilation time was associated with a cumulative dose of 560 mg of methylprednisolone (which corresponds to 105 mg of dexamethasone). However, Maia et al. did not report other important clinical outcomes, particularly mortality.

The beneficial effects of higher corticosteroid doses on clinical outcomes have been demonstrated in studies with cumulative dexamethasone doses ranging from approximately 120 to 130 mg [11, 12]. These findings are consistent with our study, which showed that there was lower mortality when the cumulative dose of dexamethasone equivalence increased from low (< 126 mg) to moderate (126.01–165.00 mg) doses. Additionally, a study by Toroghi et al. revealed a trend of increasing 60-day mortality from 17% to 30% and 41% with cumulative dexamethasone doses of 80 mg, 160 mg, and 240 mg, respectively [13]. Similarly, in our investigation, cumulative doses of dexamethasone equivalent exceeding 165 mg were associated with increased in-hospital mortality.

The strength of our study lies in the consideration of both the cumulative dose and the timing of corticosteroid therapy. We followed clinical guidelines for personalized steroid modulation on the basis of patient severity and risk factors, adjusting the daily dose in response to parameters such as the degree of hypoxemia, chest radiography progression, and C-reactive protein levels [16]. Furthermore, the complications from corticosteroid therapy [17] were carefully monitored and recorded due to concerns about potential harm.

However, our study has several limitations. First, it was a retrospective, observational investigation with a limited number of patients and several unmeasured confounders. Thus, a larger trial is needed to confirm or refute these findings. Second, since the data were collected during the pandemic, therapeutic agents such as remdesivir, tocilizumab, and baricitinib were limited in availability, potentially influencing the clinical outcomes observed in this study. Third, we used the cumulative dexamethasone equivalent dose during the treatment period, which probably did not represent the daily dose of corticosteroids in COVID-19 patients with pneumonia in real-world practice. Fourth, our population varied in terms of age, comorbidities, days since symptom onset, and severity of disease, which may have resulted in heterogeneity in the clinical outcomes. Fifth, most of the patients in this study were recruited during the delta variant pandemic of SARS-CoV-2, which is not the current strain. Therefore, the results might not apply to the current variant. Sixth, our study reported hospital mortality as the primary outcome and mortality at different time points as secondary outcomes. We did not report the actual cause of death for each patient, which could have obscured a mechanism of death that might not be related to COVID-19. Seventh, we recruited only hypoxemic patients requiring corticosteroids, and thus, we could not compare the outcomes of these patients with those of patients not receiving corticosteroids.

## Conclusions

The intensity of corticosteroid therapy, measured as the cumulative dexamethasone-equivalent dose during the first 14 days, is not a risk factor for in-hospital mortality in hypoxemic COVID-19 patients. However, cumulative doses exceeding 165 mg are associated with higher in-hospital mortality and secondary hospital-acquired infection.

## Supporting information

**S1 Checklist. TREND statement checklist.**
(PDF)

**S1 Table. Dexamethasone equivalent dose.**
(DOCX)

**S2 Table. Univariable and multivariable analyses.**
(DOCX)

**S3 Table. Demographic data, treatment and clinical outcomes stratified into quartiles according to the cumulative dexamethasone equivalent dose.**
(DOCX)

**S4 Table. Previous study.**
(DOCX)

**S1 Data.**
(XLSX)

## Acknowledgments

We would like to express our deep gratitude to Suthipol Udompunthurak for statistical analysis consultation and correction.

## Author Contributions

**Conceptualization:** Ranistha Ratanarat.

**Data curation:** Napassorn Teeratakulpisarn, Supattra Chiewroongroj.

**Formal analysis:** Napassorn Teeratakulpisarn, Thummaporn Naorungroj.

**Investigation:** Napassorn Teeratakulpisarn, Supattra Chiewroongroj.

**Methodology:** Napassorn Teeratakulpisarn, Thummaporn Naorungroj, Ranistha Ratanarat.

**Project administration:** Ranistha Ratanarat.

**Supervision:** Ranistha Ratanarat.

**Writing – original draft:** Napassorn Teeratakulpisarn.

**Writing – review & editing:** Napassorn Teeratakulpisarn, Thummaporn Naorungroj, Ranistha Ratanarat.

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
