## [Decision Letter · Decision Letter 0]

22 Apr 2024

PONE-D-24-03848Effects of Corticosteroid Intensity on Clinical Outcomes in Hypoxic COVID-19 PatientsPLOS ONE

Dear Dr. Ratanarat,

Thank you for submitting your manuscript to PLOS ONE. After careful consideration, we feel that it has merit but does not fully meet PLOS ONE’s publication criteria as it currently stands. Therefore, we invite you to submit a revised version of the manuscript that addresses the points raised during the review process.

We look forward to receiving your revised manuscript.

Kind regards,

Jiawen Deng

Academic Editor

PLOS ONE

Journal Requirements:

Reviewers' comments:

Reviewer's Responses to Questions

**Comments to the Author**

1. Is the manuscript technically sound, and do the data support the conclusions?

Reviewer #1: Partly

Reviewer #2: Partly

2. Has the statistical analysis been performed appropriately and rigorously? 

Reviewer #1: Yes

Reviewer #2: Yes

3. Have the authors made all data underlying the findings in their manuscript fully available?

Reviewer #1: Yes

Reviewer #2: No

4. Is the manuscript presented in an intelligible fashion and written in standard English?

Reviewer #1: No

Reviewer #2: No

5. Review Comments to the Author

**Reviewer #1: **Thank you for the opportunity to review the manuscript "Effects of Corticosteroid Intensity on Clinical Outcomes in Hypoxic COVID-19 Patients". The authors seek to address the knowledge gap in optimal dose and duration of steroid dosing in COVID-19 pneumonia patients using a retrospective cohort study, categorizing cumulative dexamethasone dosing over 14 days into quartiles.

Major questions to address:

- why did the authors choose to study an intervention duration of 14 days? (most of the cited studies included 10 day duration)

- why did the authors choose to study cumulative dexamethasone dosing by quartiles? (most cited studies included high intensity vs lower intensity vs placebo, high intensity dosing was usually 20mg daily for 5 days followed by 10mg daily for 5 days, total 150mg; although one study did include 8mg 3x daily for 10 days, 240 mg total)

- why was the primary outcome in-hospital mortality? (most of the cited studies used 28 day mortality, 28d VFD, and 60 day mortality)

- please check references, #2 and #3 do not have functional links

- P13 L237-238: please rewrite, "this causal relationship", impossible to determine a causal relationship with a retrospective study

Additional issues to address:

- please correct grammar, several instances of awkward wording throughout

- P4 L62-66: please update reference for pathogenesis and treatment, link does not work

- P9 L152: why was lymphocyte count used, but white blood cell count not included?

- why were the inclusion criteria limited to patients requiring oxygen therapy? (several cited studies included patients not requiring oxygen therapy)

- please edit "hospital mortality" to "in-hospital mortality" throughout

**Reviewer #2**: Thank you for the opportunity to review this manuscript. Teeratakulpisarn et al. present a retrospective study of patients with COVID-19 to determine associations between corticosteroid dosages and patient outcomes. While this is interesting, I am concerned that this study may no longer be as timely and most institutions have specific, predefined dosages at this point in time. The strains/variants have also changed considerably and it is unclear how pertinent this data would be.

Major:

1) The study is single-center, across almost 18 months only. This should be further elaborated upon in the limitations as it has a considerable impact on generalizability. Moreover, the emergence of new variants should be discussed as a limitation of the study.

2) While the primary outcome is hospital mortality and secondary outcomes are mortality at different time points, the actual causes of death are not reported. This would be important to note and if unavailable, this is a major limitation as it is unclear as to the mechanism of death.

3) Hospital-acquired infections should be defined and noted with respect to how they were identified.

4) The result section reports median (IQR) but the abstract is in mean (SD) for often the same variables (e.g. dosage). As per the methods, depending on the distribution, the authors should select one and be consistent.

5) The authors should justify and provide references for why and how these quartiles were decided upon.

6) The confidence intervals, particularly for ECMO and ventilator use in Table S1 are extremely wide and depending on the data type (e.g. large number of 0s relative to 1s), the authors should consider other techniques such as negative binomial regressions (https://stats.oarc.ucla.edu/stata/dae/negative-binomial-regression/)

Minor:

1) Title: Consider rephrasing to "The association between corticosteroid dosage and clinical outcomes in COVID-19 pneumonia: A retrospective cohort study" or something similar

2) Abstract - Conclusion: The authors should ensure all data is presented in the form of associations rather than causal given the study design. Consider changing "risk factor" to "associated with"

3) Through out the manuscript causal language is used (e.g. "increased"). This should all be reworded to "higher" or "lower," etc.

4) Key message: Please clarify what a "J-curve" really means, especially in the key message section. "Linked" should also be modified to "associated."

5) Introduction: The first paragraph, especially the first sentence should be revised to reflect the current climate of COVID-19. Specifically, I would not call it "emerging" at this point in time.

6) Introduction, Lines 79-80: The authors should consider including trials which have indeed looked at steroid dosages (e.g. https://jamanetwork.com/journals/jama/fullarticle/2785529)

7) All acronyms need to be defined (e.g. ECMO, PF, etc). Some percentages in the results are also inadvertently subscripted.

8) Tables: The number of decimal places for the data should be consistent. They would all benefit from inclusion of abbreviation definitions at the bottom of all main and supplementary tables

9) The study would benefit from an English grammar review

6. PLOS authors have the option to publish the peer review history of their article (what does this mean?). If published, this will include your full peer review and any attached files.

Reviewer #1: No

Reviewer #2: No

---

## [Author Response · Author response to Decision Letter 0]

29 May 2024

Professor Jiawen Deng

Editor-in-Chief,

PLOS ONE Journal

May 25th, 2024

Answer to review of: PONE-D-24-03848 ‘Effects of Corticosteroid Intensity on Clinical Outcomes in Hypoxic COVID-19 Patients

Professor Jiawen Deng

 Thank you for your potential interest in publishing our work. We appreciate the valuable comments from the reviewers and believe that their suggestion improved our work. We have revised and added information to our manuscript in response to the reviewers’ comments, and below we provide responses to each issue raised.

 All comments are restated below followed by our responses and description of the substance and location of any resulting changes made to the revised manuscript.

Yours sincerely

Napassorn Teeratakulpisarn

Thummaporn Naorungroj

Asso. Prof. Ranistha Ratanarat

Department of Intensive Care 

Faculty of Medicine Siriraj Hospital Mahidol University

 

Response to Reviewers

RESPONSE: Thank you for your recommendation. We informed the reviewer that this study is a retrospective study. Thus, we haven’t published a preplanned laboratory research protocol. However, we have provided our data in supporting files.

Reviewer #1: 

- Why did the authors choose to study an intervention duration of 14 days? (most of the cited studies included 10 days duration)

RESPONSE: We thank the reviewer for this comment. We informed the reviewer that the patients including in this study were received the treatments during the pandemic era of the delta variant before the publication of RECOVRY or REMAP-CAP study. In our institute, we have followed the local guideline adopted from the WHO guideline for corticosteroid tailoring based on individual patient characteristics, severity, and risk factors (1). Moreover, 71.2% of the patients in this study receiving one of systemic corticosteroids for more than 10 days. Hence, we decided to present corticosteroid dosing in a cumulative dexamethasone dose over 14 days.

Reference

1 Ratanarat R, Thitayanapong A. Rational use of corticosteroid treatment in the early phase of severe COVID-19: Corticosteroid in COVID-19. Clin Crit Care [Internet]. 2023 Jun. 26 [cited 2024 May 5];31(1):2023:e0010. 

- Why did the authors choose to study cumulative dexamethasone dosing by quartiles? (most cited studies included high intensity vs lower intensity vs placebo, high intensity dosing was usually 20mg daily for 5 days followed by 10mg daily for 5 days, total 150mg; although one study did include 8mg 3x daily for 10 days, 240 mg total)

RESPONSE: We thank the reviewer for this thoughtful comment. There were some reasons why we have chosen to study cumulative dexamethasone dosing by quartiles. First, the characteristics of our patients were more severe COVID-19 compared to most cited studies (1-4) (Supplementary table S4), with a median PF ratio of 151.4 (IQR: 104.8, 224.0) and 53.5% of patients requiring mechanical ventilator support. In our cohort, the median dose of corticosteroid was 165 mg which was exceeding high-intensity dose using in CoDEX trial (5). Furthermore, we did not enroll the patient who did not receive corticosteroid so we did not have placebo group. Second, this study aimed to observe a correlation between the optimal dose (neither too low nor too high) and favorable clinical outcomes in our center, as well as in a previous study conducted by Maia et al (6). Third, we, indeed, have conducted an analysis comparing high vs low intensity dosing, and the results showed there was no significant difference in mortality. However, we hypothesized that the cumulative dose of corticosteroids may not a linear correlation. Therefore, we have attempted to present the corticosteroid dosing in more granular data as shown in this study. Fourth, in the statistical view, dividing cumulative dexamethasone dosing into quartiles would give equal data for each group and it is concordance with box plot analysis.

Reference

1 Durr KM, Hendin A, Perry JJ. Effect of 12 mg vs 6 mg of dexamethasone on the number of days alive without life support in adults with COVID-19 and severe hypoxemia: the COVID STEROID 2 randomized trial. CJEM. 2022 Apr;24(3):266-267.

2 Maskin LP, Bonelli I, Olarte GL, Palizas F Jr, Velo AE, Lurbet MF, et al. High- Versus Low-Dose Dexamethasone for the Treatment of COVID-19-Related Acute Respiratory Distress Syndrome: A Multicenter, Randomized Open-Label Clinical Trial. J Intensive Care Med. 2022 Apr;37(4):491-499.

3 Toroghi N, Abbasian L, Nourian A, Davoudi-Monfared E, Khalili H, Hasannezhad M, et al. Comparing efficacy and safety of different doses of dexamethasone in the treatment of COVID-19: a three-arm randomized clinical trial. Pharmacol Rep. 2022 Feb;74(1):229-240.

4 Bouaddma L, Mekontso-Dessap A, Burdet C, Merdji H, Poissy J, Dupuis C, Guitton C, Schwebel C, Cohen Y, Bruel C, Marzouk M, Geri G, Cerf C, Megarbane B, Garcon P, Kipnis E, Visseaux B, Beldjoudi N, Chevret S, Timsit JF; COVIDICUS Study Group. High-Dose Dexamethasone And Oxygen Support Strategies in Intensive Care Unit Patients With Severe COVID-19 Acute Hypoxemic Respiratory Failure: The COVIDICUS Randomized Clinical Trial. JAMA Intern Med. 2022 Sep 1;182(9):906-916.

5 Tomazini BM, Maia IS, Cavalcanti AB, Berwanger O, Rosa RG, Veiga VC, et al. Effect of Dexamethasone on Days Alive and Ventilator-Free in Patients With Moderate or Severe Acute Respiratory Distress Syndrome and COVID-19: The CoDEX Randomized Clinical Trial. JAMA. 2020 Oct 6;324(13):1307-1316.

6 Maia R, Melo L, Mendes JJ, Freitas PT. Corticosteroids in COVID-19: A double-edged sword—a retrospective study. Med Intensiva (Engl Ed). 2022 Apr;46(4):229-231.

- Why was the primary outcome in-hospital mortality? (most of the cited studies used 28-day mortality, 28d VFD, and 60-day mortality)

RESPONSE: We thank the reviewer for your comment. We have to use the in-hospital mortality instead of 28-day mortality for primary outcome because according to retrospective data during the pandemic, there was a significant missing data for the 28-day mortality data. Therefore, we have selected in-hospital mortality, which was the most complete mortality outcome, as the primary outcome.

- Please check references, #2 and #3 do not have functional links

RESPONSE: We apologized the reviewer for these errors. We have already corrected the functional links for reference 2 and 3 in our revised manuscript.

- P13 L237-238: please rewrite, "this causal relationship", impossible to determine a causal relationship with a retrospective study

RESPONSE: We thank the reviewer for this comment. We have already changed “this causal relationship” to “this correlation” in our revised manuscript on page 16

- Please correct grammar, several instances of awkward wording throughout

RESPONSE: We thank the reviewer for this comment. We have rechecked the grammar and made it more appropriately.

- P4 L62-66: please update reference for pathogenesis and treatment, link does not work

RESPONSE: We thank the reviewer for this comment and We apologized for the error. We have already updated the reference for pathogenesis and treatment of COVID-19 and also corrected the functional links in our revised manuscript. 

- P9 L152: why was lymphocyte count used, but white blood cell count not included?

RESPONSE: We thank the reviewer for your thoughtful comment. We used the lymphocyte count rather than white blood cell counts because there was a significant association between low lymphocyte count and higher severity. 

References:

1 Hedayati-Ch M, Ebrahim-Saraie HS, Bakhshi A. Clinical and immunological comparison of COVID-19 disease between critical and non-critical courses: a systematic review and meta-analysis. Front Immunol. 2024;15:1341168. Published 2024 Apr 16.

2 Montiel-Cervantes LA, Medina G, Pilar Cruz-Domínguez M, et al. Poor Survival in COVID-19 Associated with Lymphopenia and Higher Neutrophile-Lymphocyte Ratio. Isr Med Assoc J. 2021;23(3):153-159.

- Why were the inclusion criteria limited to patients requiring oxygen therapy? (several cited studies included patients not requiring oxygen therapy)

RESPONSE: According to the COVID-19 treatment guidelines, systemic corticosteroids should be administered to patients requiring oxygen therapy. Therefore, we recruited only patients who receiving oxygen therapy for our study.

Reference: 

1 NIH. COVID-19 Treatment Guidelines. Clinical Management Summary. Last Updated: February 29, 2024. Available at: 

https://www.covid19treatmentguidelines.nih.gov/management/clinical-management-of-adults/clinical-management-of-adults-summary/. 

- Please edit "hospital mortality" to "in-hospital mortality" throughout

RESPONSE: We thank the reviewer for the comment. We have already changed "hospital mortality" to "in-hospital mortality" in our revised manuscript. 

Reviewer #2: 

Major:

1) The study is single-center, across almost 18 months only. This should be further elaborated upon in the limitations as it has a considerable impact on generalizability. Moreover, the emergence of new variants should be discussed as a limitation of the study.

RESPONSE: We thank the reviewer for this comment. We have already added this information in limitation in our revised manuscript as following: 

“The patients in the study were recruited only in our institute in the specific time period which was in the delta variant pandemic of SARS-CoV-2. Thus, these results were limited on generalizability and may be inapplicable to the current variant.”

2) While the primary outcome is hospital mortality and secondary outcomes are mortality at different time points, the actual causes of death are not reported. This would be important to note and if unavailable, this is a major limitation as it is unclear as to the mechanism of death.

RESPONSE: We thank the reviewer for these comments and we agreed with the reviewer that this is an important issue. We did not report the actual cause of death for each patient. Thus, we have added this information in limitation of our revised manuscript as following

 “Our study reported hospital mortality as the primary outcome and mortality at different time points as secondary outcomes. We did not report the actual cause of death for each patient, which could result in an unknown mechanism of death that might not be related to COVID-19.”

3) Hospital-acquired infections should be defined and noted with respect to how they were identified.

RESPONSE: We thank the reviewer for this comment. We have added the definition of “Hospital-acquired infection” in method at P7 L 122-124 as the following

 “hospital-acquired infections were including hospital acquired pneumonia, urinary tract infection, bacteremia, catheter related bloodstream infection, and skin and soft tissue infection”.

4) The result section reports median (IQR) but the abstract is in mean (SD) for often the same variables (e.g. dosage). As per the methods, depending on the distribution, the authors should select one and be consistent.

RESPONSE: We thank the reviewer for these comments and we apologized for these errors. We have already corrected the results and have reported the median (interquartile range) according to the distribution for every section in the manuscript.

5) The authors should justify and provide references for why and how these quartiles were decided upon.

RESPONSE: RESPONSE: We thank the reviewer for this thoughtful comment. There were some reasons why we have chosen to study cumulative dexamethasone dosing by quartiles. First, the characteristics of our patients were more severe COVID-19 compared to most cited studies (1-4) (Supplementary table S4), with a median PF ratio of 151.4 (IQR: 104.8, 224.0) and 53.5% of patients requiring mechanical ventilator support. In our cohort, the median dose of corticosteroid was 165 mg which was exceeding high-intensity dose using in CoDEX (5). Furthermore, we did not enroll the patient who did not receive corticosteroid so we did not have placebo group. Second, this study aimed to observe a correlation between the optimal dose (neither too low nor too high) and favorable clinical outcomes in our center, as well as in a previous study conducted by Maia et al (6). Third, we, indeed, have conducted an analysis comparing high vs low intensity dosing, and the results showed there was no significant difference in mortality. However, we hypothesized that the cumulative dose of corticosteroids may not a linear correlation. Therefore, we have attempted to present the corticosteroid dosing in more granular data as shown in this study. Fourth, in the statistical view, dividing cumulative dexamethasone dosing into quartiles would give equal data for each group and it is concordance with box plot analysis.

Reference

1 Durr KM, Hendin A, Perry JJ. Effect of 12 mg vs 6 mg of dexamethasone on the number of days alive without life support in adults with COVID-19 and severe hypoxemia: the COVID STEROID 2 randomized trial. CJEM. 2022 Apr;24(3):266-267.

2 Maskin LP, Bonelli I, Olarte GL, Palizas F Jr, Velo AE, Lurbet MF, et al. High- Versus Low-Dose Dexamethasone for the Treatment of COVID-19-Related Acute Respiratory Distress Syndrome: A Multicenter, Randomized Open-Label Clinical Trial. J Intensive Care Med. 2022 Apr;37(4):491-499.

3 Toroghi N, Abbasian L, Nourian A, Davoudi-Monfared E, Khalili H, Hasannezhad M, et al. Comparing efficacy and safety of different doses of dexamethasone in the treatment of COVID-19: a three-arm randomized clinical trial. Pharmacol Rep. 2022 Feb;74(1):229-240.

4 Bouaddma L, Mekontso-Dessap A, Burdet C, Merdji H, Poissy J, Dupuis C, Guitton C, Schwebel C, Cohen Y, Bruel C, Marzouk M, Geri G, Cerf C, Megarbane B, Garcon P, Kipnis E, Visseaux B, Beldjoudi N, Chevret S, Timsit JF; COVIDICUS Study Group. High-Dose Dexamethasone And Oxygen Support Strategies in Intensive Care Unit Patients With Severe COVID-19 Acute Hypoxemic Respiratory Failure: The COVIDICUS Randomized Clinical Trial. JAMA Intern Med. 2022 Sep 1;182(9):906-916.

5 Tomazini BM, Maia IS, Cavalcanti AB, Berwanger O, Rosa RG, Veiga VC, et al. Effect of Dexamethasone on Days Alive and Ventilator-Free in Patients With Moderate or Severe Acute Respiratory Distress Syndrome and COVID-19: The CoDEX Randomized Clinical Trial. JAMA. 2020 Oct 6;324(13):1307-1316.

6 Maia R, Melo L, Mendes JJ, Freitas PT. Corticosteroids in COVID-19: A double-edged sword—a retrospective study. Med Intensiva (Engl Ed). 2022 Apr;46(4):229-231.

6) The confidence intervals, particularly for ECMO and ventilator use in Table S1 are extremely wide and depending on the data type (e.g. large number of 0s relative to 1s), the authors should consider other techniques such as negative binomial regressions (https://stats.oarc.ucla.edu/stata/dae/negative-binomial-regression/)

RESPONSE: We thank the reviewer for your thoughtful comment. We have discussed our data with statistician and we have determined that the negative binomial regression is typically used for modeling count variables, usually for over-dispersed count outcome variables. However, since our data uses hospital mortality as an outcome, which is a binary outcome, we have opted to use binomial logistic regression for the analysis.

References:

1 Statistical Method and Data Analytics [Internet]. UCLA. LOGISTIC REGRESSION | STATA DATA ANALYSIS EXAMPLES [cited 2024 May 12]. Available from:

https://stats.oarc.ucla.edu/stata/dae/logistic-regression/

2 Statistical Method and Data Analytics [Internet]. UCLA. NEGATIVE BINOMIAL REGRESSION | STATA DATA ANALYSIS EXAMPLES [cited 2024 May 12]. Available from:

https://stats.oarc.ucla.edu/stata/dae/negative-binomial-regression/

---

## [Decision Letter · Decision Letter 1]

19 Jun 2024

PONE-D-24-03848R1The association between corticosteroid dosage and clinical outcomes in COVID-19 pneumonia: A retrospective cohort studyPLOS ONE

Dear Dr. Ratanarat,

Thank you for submitting your revised manuscript to PLOS ONE. While the reviewers no longer have major comments on the revised article, they raised a few minor points that need to be addressed. Specifically, I agree with the reviewers that your manuscript could benefit from English editing.

We look forward to receiving your revised manuscript.

Kind regards,

Jiawen Deng

Academic Editor

PLOS ONE

Journal Requirements:

Reviewers' comments:

Reviewer's Responses to Questions

**Comments to the Author**

1. If the authors have adequately addressed your comments raised in a previous round of review and you feel that this manuscript is now acceptable for publication, you may indicate that here to bypass the “Comments to the Author” section, enter your conflict of interest statement in the “Confidential to Editor” section, and submit your "Accept" recommendation.

Reviewer #1: All comments have been addressed

Reviewer #2: All comments have been addressed

2. Is the manuscript technically sound, and do the data support the conclusions?

Reviewer #1: Partly

Reviewer #2: Yes

3. Has the statistical analysis been performed appropriately and rigorously? 

Reviewer #1: I Don't Know

Reviewer #2: Yes

4. Have the authors made all data underlying the findings in their manuscript fully available?

Reviewer #1: Yes

Reviewer #2: Yes

5. Is the manuscript presented in an intelligible fashion and written in standard English?

Reviewer #1: Yes

Reviewer #2: Yes

6. Review Comments to the Author

Reviewer #1: Thank you for the opportunity to review the revised manuscript "The association between corticosteroid dosage and clinical outcomes in COVID-19 pneumonia: A retrospective cohort study"

- if the patient population factored into the decision to study dexamethasone quartiles, the title should reflect that this is a study in severe, hypoxemic COVID-19 pneumonia

- having no patients receiving corticosteroid or no placebo group should be mentioned in the limitations section

- while greatly improved, there are still many instances where the wording is awkward or confusing, while I do not have a specific institute/agency recommendation in mind, this manuscript would benefit from further English grammar review

Reviewer #2: I thank the authors for their careful consideration of my comments and their thoughtful replies. I do not have additional recommendations given this thorough revision.

7. PLOS authors have the option to publish the peer review history of their article (what does this mean?). If published, this will include your full peer review and any attached files.

Reviewer #1: No

Reviewer #2: No

---

## [Author Response · Author response to Decision Letter 1]

11 Jul 2024

Reviewer #1: 

- If the patient population factored into the decision to study dexamethasone quartiles, the title should reflected thst this is a study in severe, hypoxemic COVID-19 pneumonia

RESPONSE: We thank the reviewer for this comment. We have changed the title to “Associations between corticosteroid dosage and clinical outcomes in patients with hypoxemic COVID-19 pneumonia: A retrospective cohort study”

- having no patients receiving corticosteroid or no placebo group should be mentioned in the limitations section

RESPONSE: We thank the reviewer for this thoughtful comment. We added the seventh limitation in P18, L 289-292 that “we recruited only hypoxemic patients requiring corticosteroids, and thus, we could not compare the outcomes of these patients with those of patients not receiving corticosteroids.”

- while greatly improved, there are still many instances where the wording is awkward or confusing, while I do not have a specific institute/agency recommendation in mind, this manuscript would benefit from further English grammar review

RESPONSE: We thank the reviewer for your comment. We have sent our manuscript to AJE academic English editing services for a grammar review and have corrected our manuscript according to their suggestions.

---

## [Editor Report · Decision Letter 2]

17 Jul 2024

Associations between corticosteroid dosage and clinical outcomes in patients with hypoxemic COVID-19 pneumonia: A retrospective cohort study

PONE-D-24-03848R2

Dear Dr. Ratanarat,

Your manuscript has improved significantly over the last two rounds of revisions. As such, we can now accept your submission for publication in PLOS ONE. Thank you for your insightful submission.

Kind regards,

Jiawen Deng

Academic Editor

PLOS ONE
---

## [Editor Report · Acceptance letter]

26 Jul 2024

PONE-D-24-03848R2 

PLOS ONE

Dear Dr. Ratanarat, 

I'm pleased to inform you that your manuscript has been deemed suitable for publication in PLOS ONE. Congratulations! Your manuscript is now being handed over to our production team.

Kind regards, 

on behalf of

Dr. Jiawen Deng 

Academic Editor

PLOS ONE